

# Northern Greenland transect stacked ice cores as a proxy for winter extreme events in Europe

Alessandro Gagliardi[1], Norel Rimbu[1], Gerrit Lohmann[1,2], and Monica Ionita[1,3]

[1]Paleoclimate Dynamics Group, Alfred Wegener Institute, Helmholtz Centre for Polar and Marine Research, Bremerhaven, Germany
[2]Physics Department, University of Bremen, Bremen, Germany
[3]Forest Biometrics Laboratory – Faculty of Forestry, Ştefan cel Mare University of Suceava, Suceava, Romania

**Correspondence:** Alessandro Gagliardi (alessandro.gagliardi@awi.de)

**Abstract.**

High-resolution ice core records from the Greenland ice sheet provide critical insights into past climate variability across seasonal to multidecadal timescales. A key proxy in these reconstructions is the concentration of stable oxygen isotopes ($\delta^{18}$O), which reflects both regional climatic conditions, such as temperature, as well as atmospheric and oceanic circulation patterns.

While recent studies have linked $\delta^{18}$O variability to synoptic-scale phenomena, particularly atmospheric blocking, its relationship to extreme hydroclimatic events in Europe remains underexplored. This study demonstrates that a stacked record of $\delta^{18}$O from the Northern Greenland Transect (NGT), spanning 1602 to 2011, serves as a proxy for hydroclimatic extremes in Europe. The connection between $\delta^{18}$O anomalies and European atmospheric circulation patterns is investigated across two periods: the observational era (1920–2011) and a longer historical context (1602–2003) using paleoclimate reanalysis data. Composite

analysis reveals that years characterized by low $\delta^{18}$O values in the NGT record correspond to an increased frequency of atmospheric blocking over Europe. These blocking events are associated with distinct hydroclimatic extremes. Specifically, the analysis shows a consistent pattern of enhanced frequency of extreme precipitation along Norwegian coast and more frequent extreme drier conditions over southern Europe during such years. The persistence of this linkage in both modern observations and long-term reconstructions underscores the robustness and temporal stability of the relationship between Greenland $\delta^{18}$O

variability and European hydroclimatic extremes driven by atmospheric blocking.

## 1   Introduction

The frequency of extreme weather events is rising due to climate change, leading to significant loss of life and substantial socio-economic costs (Faranda et al., 2023b; Bakke et al., 2023; Ma et al., 2024; Ionita and Nagavciuc, 2025). These events typically last from a few days (e.g. heatwaves, cold spells) up to months (e.g., droughts). In recent years, Europe has experienced

an increasing frequency of extreme weather events (e.g. IPCC, 2021; Vautard et al., 2023; Dong and Sutton, 2025). Several factors contribute to this trend. Scaife et al. (2008) linked the rise in extreme events to long-term changes in the North Atlantic Oscillation (NAO), while García-Burgos et al. (2023) attributed it to a combination of jet stream dynamics and atmospheric blockings. Atmospheric blocking plays a key role in extreme weather by disrupting the jet stream, creating meanders that foster





conditions conducive to severe events (Kautz et al., 2022). Additionally, Faranda et al. (2023a) found that certain atmospheric
circulation patterns are becoming more frequent, increasing the recurrence of extreme events, while others are declining,
leading to fewer corresponding extremes. Kornhuber and Messori (2023) also demonstrated that high-amplitude wave-4 events
have become more common across the Northern Hemisphere, enhancing the likelihood of extreme weather in both North
America and Europe. When such extreme events persist over a region for extended periods, they can be classified as extreme
climate events. Understanding the past evolution of extreme weather events is crucial for investigating changes in underlying
physical processes, validating climate models, and societal impacts.

However, the lack of high temporal resolution in proxies data makes a challenge reconstructing weather extreme events.
There exist some proxies able to provide a daily-frequency resolution, but they are limited in capturing regional patterns (Yan
et al., 2020). Recently, a network of tree rings data together with deep learning techniques has been used to reconstruct summer
atmospheric blocking for the last millennia (Karamperidou, 2024). On the other hand, the time span covered by tree-ring
reconstructions is limited compared to other natural archives, such as ice cores.

Ice cores records can be used for multidecadal and longer time scale reconstructions (Rimbu and Lohmann, 2010b). The
growing number of high temporal resolution ice cores from the Greenland ice sheet gives valuable information on climate
variations from seasonal to multidecadal time scales. The variability of $\delta^{18}$O from Greenland ice cores is linked to changes
in the general climate and associated teleconnection patterns such as NAO or the Atlantic Multidecadal Oscillation (AMO)
(Barlow et al., 1993; Appenzeller et al., 1998; Vinther et al., 2003b; Chylek et al., 2011). Recent studies, though, have identified
strong links between Greenland $\delta^{18}$O variability and atmospheric weather regimes (Rimbu and Lohmann, 2010a; Ortega et al.,
2014) and relationship with atmospheric blocking during boreal winter months (Rimbu et al., 2007, 2017, 2021).

The primary objective is to investigate whether a connection between $\delta^{18}$O variability measured in ice cores and extreme
climate events exist during the boreal winter. To this end, this paper assesses the validity of the $\delta^{18}$O variability in the Northern
Greenland Transect (NGT) stacked ice cores (Hörhold et al., 2023) is a proxy for extreme climate events.

The paper is structured as follows: Sect. 2 outlines the datasets and methods used for investigation the atmospheric circulation
and the associated extreme climate events, while Sect. 3 describes the results obtained for the observational and long-term
perspective periods. Lastly, Sect. 4 summaries the results.

## 2 Data and methods

### 2.1 Data

The proxy data used is the NGT stacked stable oxygen isotope record $\delta^{18}$O obtained from several ice cores drilled mainly from
the Northern Greenland (Hörhold et al., 2023). The NGT stacked $\delta^{18}$O series is a stacked record of many different $\delta^{18}$O ice
cores measurements across different time spans. Stacking measurements from different drilling sites has the major advantage of
averaging out different local processes affecting on the $\delta^{18}$O variability. The NGT stacked $\delta^{18}$O series is provided as anomalies
with respect to period $1961 - 1990$. For details on the stacking method and data sources see Hörhold et al. (2023).



The atmospheric circulation during boreal winters in the observational period $(1920 - 2011)$ is investigated using the third version of the 20th Century reanalysis (20CRv3) (Slivinski et al., 2019), which provides daily data and a spatial resolution of $1.0° \times 1.0°$. In particular, the atmospheric variables used in this study are the geopotential height at 500hPa, $Z_{500}$, along with the zonal $u$ and meridional $v$ components of the wind at the same pressure level. The specific humidity $q$, the zonal, and meridional components of the wind at different pressure levels (300hPa to 1000hPa) are used to investigate the water vapor transported by the atmosphere. The pressure levels are limited between 300hPa and 1000hPa because most water vapor is in the lower troposphere. In particular, the water vapor transport is determined by the Integrated Vapor Transport (IVT) (Peixoto and Oort, 1992),

$$\mathrm{IVT} = \sqrt{\left(\frac{1}{g}\int\limits_{1000}^{300} qu dp\right)^2 + \left(\frac{1}{g}\int\limits_{1000}^{300} qv dp\right)^2}, \tag{1}$$

where $g$ is the gravitational acceleration in $\mathrm{m\,s^{-2}}$, $u$ and $v$ are the zonal and meridional components of the wind in $\mathrm{m\,s^{-1}}$, $q$ the specific humidity in $\mathrm{kg\,kg^{-1}}$ and $dp$ is the pressure difference between two adjacent pressure levels.

The extreme conditions regarding temperature and precipitation over Europe in boreal winters are assessed in the observational period using the E-OBS dataset (Cornes et al., 2018). In particular, two ETCCDI extreme indices are considered: TN10p and PRCPTOT. The TN10p index is defined as the percentage of days when the minimum temperature is below the 10th percentile over a specific period, while the PRCPTOT is the cumulated rainfall in days when at least $1\,\mathrm{mm}$ of rainfall occurred. The two extreme indices are computed using the Python package icclim (Aoun et al., 2024). A comparison is also made with a newly reconstruction Hadex dataset using artificial intelligence named CRAI (Plésiat et al., 2024). This dataset provides a reconstructions of some extreme indices, including TN10p, using a deep-learning algorithm that combines U-net architecture and partial convolutional layers (Kadow et al., 2020).

The Ensemble Kalman Filter (EKF) v2 Paleoreanalysis (Valler et al., 2022) is used to investigate atmospheric circulation patterns, temperature and precipitation in the boreal winters in the long-term perspective period $(1602 - 2003)$. The EKF dataset has monthly time resolution and spatial resolution of $1.85° \times 1.85°$. This data set provides an optimized combination of model outputs and paleoclimate reconstructions obtained by different proxies through data assimilation techniques. See Valler et al. (2022) for details on the data assimilation techniques and proxies adopted.

## 2.2 Methods

The principal method of investigation in this paper is the composite map analysis (DelSole and Tippett, 2022). The aim of the composite analysis is to identify relations between two physical variables during the occurrence of particular extreme conditions in of the two variables. In this case, the composite analysis is used to identify the average atmospheric circulation patterns when the NGT stacked $\delta^{18}$O series assumes particularly high and low values according to certain thresholds. The choice of these threshold values is a trade-off between capturing years when $\delta^{18}$O series takes extreme values and having a congruous number of observations available to carry out the composite analysis. The choice of $\pm 1\sigma$ meets both criteria, where $\sigma$ is the standard deviation of the NGT stacked $\delta^{18}$O series during the period considered. It follows that positive years are





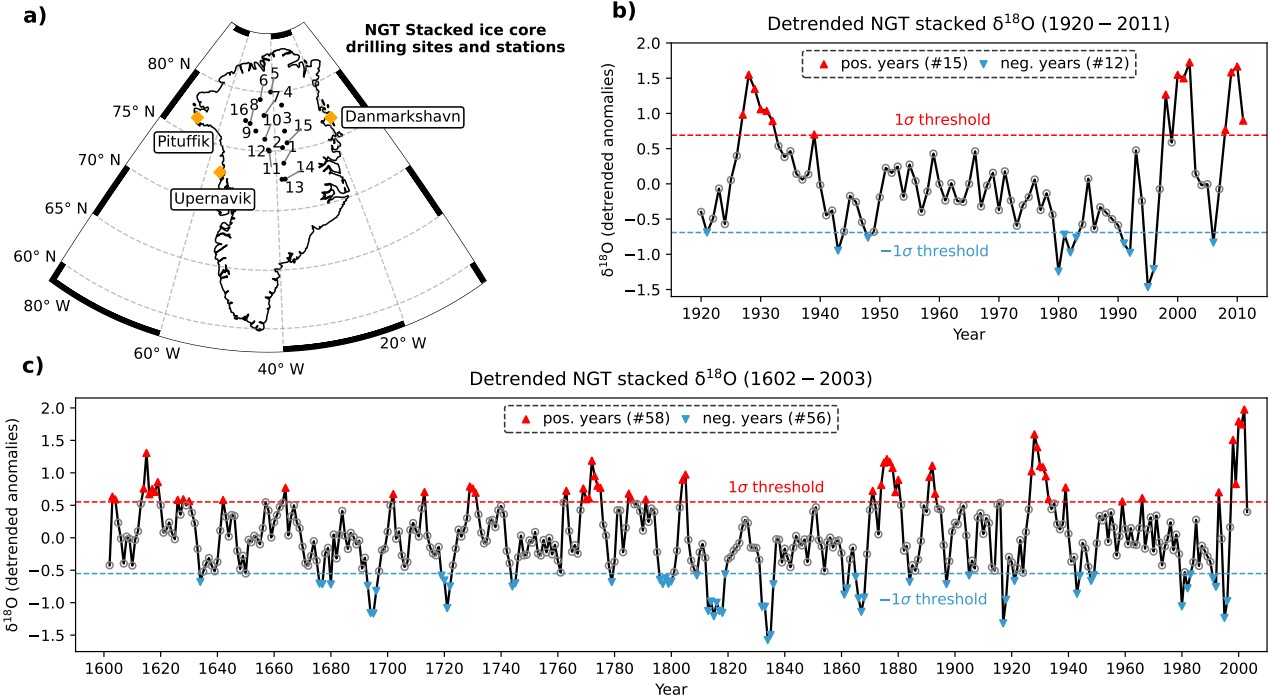

**Figure 1.** Ice cores locations (dots) and weather stations (diamonds) used in NGT Stacked ice cores **(a)**. The detrended anomalies of NGT stacked $\delta^{18}O$ series over the period $1920-2011$ and $1602-2003$ **(b, c)**. The upward-pointing red triangles refer to years whenever the $\delta^{18}O$ is above the threshold $1\sigma$ (pos. years) ($\sigma$ = standard deviation). Similarly, downward-pointing blue triangles refer to years below the threshold $-1\sigma$ (neg. years).

whenever the NGT stacked $\delta^{18}O$ series assumes values larger than $1\sigma$. Similarly, negative years are whenever $\delta^{18}O$ series is lower than the $-1\sigma$. Throughout the paper, the years where $\delta^{18}O$ series takes values above (below) the thresholds are named

positive (negative) years.

To assess the presence of atmospheric blocking, the 1D- Tibaldi and Molteni (1990, hereafter TM90) and 2D-index (Davini et al., 2012, hereafter ED12) atmospheric blocking are computed with the MiLES software (Davini, 2019). A description of the ED12 index is provided below. For details on TM90 index see Tibaldi and Molteni (1990).

For each grid point at longitude $\lambda_0$ and latitude $\Phi_0$ (from $30°$ N to $75°$ N), the geopotential height gradients GHGS and

GHGN are computed as:

$$\mathrm{GHGS}(\lambda_0, \Phi_0) = \frac{Z_{500}(\lambda_0, \Phi_0) - Z_{500}(\lambda_0, \Phi_S)}{\Phi_0 - \Phi_S}, \tag{2}$$

$$\mathrm{GHGN}(\lambda_0, \Phi_0) = \frac{Z_{500}(\lambda_0, \Phi_N) - Z_{500}(\lambda_0, \Phi_0)}{\Phi_N - \Phi_0}, \tag{3}$$

where $\Phi_N = \Phi_0 + 15°$ and $\Phi_S = \Phi_0 - 15°$. A grid point located at longitude $\lambda_0$ and latitude $\Phi_0$ is labeled as blocked if $\mathrm{GHGS}(\lambda_0, \Phi_0) > 0$ and $\mathrm{GHGS}(\lambda_0, \Phi_0) < -10\mathrm{m}(°\mathrm{lat})^{-1}$. In order to detect areas of large-scale blocking further requirements



are needed. It is then required that the grid points within 15 contiguous degrees longitude must be blocked together, along with any grid points marked as blocked in a box of 10 degrees longitude by 5 degrees latitude, for at least five days.

To asses the effect of the large-scale atmospheric circulation on boreal winter temperature and precipitation in the long-term perspective, spatial index of the geopotential height at 500hPa, average temperature and precipitation are computed, weighting each grid point by $\sqrt{\cos(\Phi)}$, where $\Phi$ is the latitude. The spatial indices are then standardized for a better comparison. A

probability density function is estimated using the non-parametric kernel estimator for identifying changes in the location, scale or skew in the distribution. Along with the probability density function, the 90% confidence intervals are computed using the simple resampling with replacement bootstrap technique. The simple replacement can be used in this application because there is no serial correlation between monthly data.

All daily atmospheric variables were averaged to derive their corresponding monthly values. A linear detrend is performed

by each month and each grid point to remove the trend and seasonality. The NGT stacked $\delta^{18}$O series has also been detrended, but each single detrending has been carried out for the observational and long-term perspective periods.

## 3   Results

The ice core locations forming the NGT stacked used for the analysis in both observational $(1920 - 2011)$ and long-term perspective $(1602 - 2003)$ periods are shown in Fig. 1a. The $\delta^{18}$O time series in the observational period (Fig. 1b) shows one

main set of extremely consecutive positive years between 1927 and 1932. Consecutive extreme positive years are also observed in the beginning of the 2000 and towards the end of the series. The consecutive extreme negative years are more evenly spread along the period, but mostly from 1980 onward. The $\delta^{18}$O series from a long-term perspective period (Fig. 1c) exhibits a well spread succession of consecutive extreme positive and negative years, even though it is often observed at least two extreme consecutive years. See Table 1 for the list of extremely positive and negative years.

The two sets of positive and negative years presented in Table 1 were identified by detrending the NGT stacked $\delta^{18}$O time series over both the observational and long-term perspective periods. Detrending the NGT stacked $\delta^{18}$O time series separately for these periods may lead to different threshold values, which could result in different identification of positive and negative years. However, the similarity between the two sets suggests that detrending over these distinct periods has little impact on the positive and negative years identification. Furthermore, the linear detreneding does not affect much the main pattern.

## 3.1   Observational period $(1920 - 2011)$

The average atmospheric circulation over Europe during positive years shows no well-defined anomaly pattern (Fig. 2a), whereas during negative years, a more pronounced pattern emerges (Fig. 2b). The first aspect noticeable is the lack of symmetry between the average atmospheric circulation patterns during the positive and negative years. The average pattern in negative years features a high-pressure system extending from the Azores Islands to the Baltic Sea and low-pressure system over

Greenland, whereas the average circulation pattern in positive years is not close to be the opposite of that in negative years. Nonetheless this asymmetry, the atmospheric circulation in positive and negative years is consistent with the wind anomaly





| Period | Type | Sub-period | Year |
|---|---|---|---|
| Observational $(1902 - 2011)$ | pos. years | $1920 - 2011$ | 1927-32, 1939, 1998, 2000-02, 2008-11 |
| | neg. years | $1920 - 2011$ | 1921, 1943, 1948, 1980-83, 1991-92, 1995-96, 2006 |
| Long-perspective $(1602 - 2003)$ | | $1602 - 1699$ | 1603-04, 1614-19, 1626, 1628, 1630, 1642, 1664 |
| | pos. years | $1700 - 1799$ | 1702, 1713, 1729-31, 1763, 1769, 1770-75, 1785, 1786, 1791 |
| | | $1800 - 1899$ | 1804-05, 1871, 1874-93 |
| | | $1900 - 2003$ | 1927-33, 1939, 1959, 1966, 1993, 1998-99, 2000-02 |
| | neg. years | $1600 - 1699$ | 1634, 1676, 1677, 1680, 1693-96 |
| | | $1700 - 1799$ | 1719-22, 1744-45, 1779, 1796, 1797, 1798, 1799, 1800 |
| | | $1800 - 1899$ | 1800, 1809, 1813-19, 1832-36, 1861-62, 1865-68, 1884, 1897 |
| | | $1900 - 2003$ | 1905, 1917-18, 1921, 1943-44, 1948-49, 1980, 1982-83, 1991-92, 1995-96 |

**Table 1.** List of all the positive (pos.) and negative (neg.) years in the observational $(1902 - 2011)$ and long-term perspective $(1602 - 2003)$ periods whenever $\delta^{18}$O is above and below the $\pm 1\sigma$ threshold ($\sigma =$standard deviation).

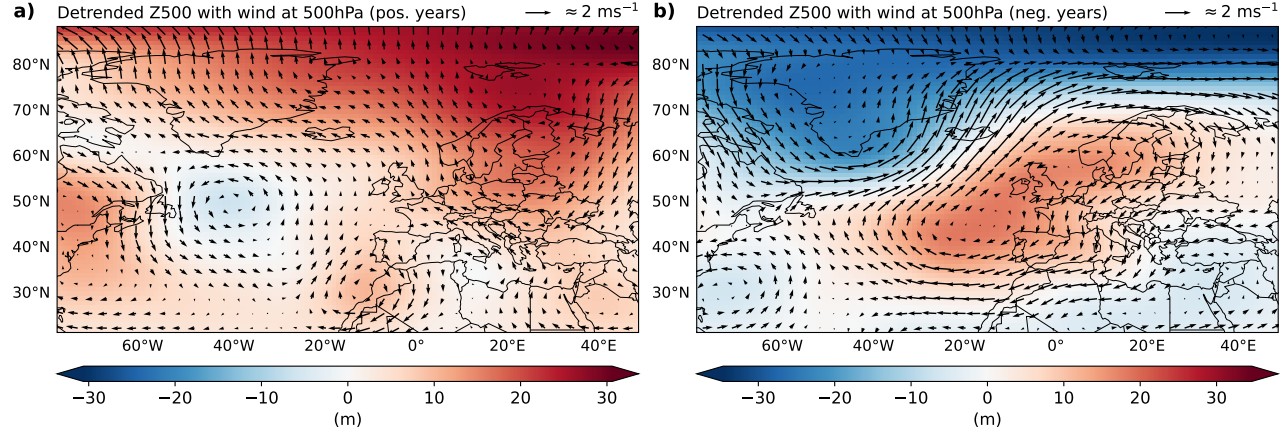

**Figure 2.** The composite maps of the detrended anomalies of the geopotential height (shaded) and the wind (vector) at 500hPa (Z500) in DJF on the positive **(a)** and negative years **(b)** of the NGT stacked ice core record for the observational period $(1920 - 2011)$.

patterns. The integrated vapor transport pattern observed during the negative years is similar to that reported by Wei et al. (2023) for particularly wet years in Southern Greenland.

The high-pressure system over Europe pattern emerging over Europe in the negative years is further investigated from a more dynamic point of view, namely in relation to atmospheric blocking. For this, both a 1D and a 2D blocking index are used. The TM90 and ED12 indices associated with negative $\delta^{18}$O years show an increase in frequency of atmospheric blocking occurring over Europe (Fig. 3a) compared to positive years. Specifically, the TM90 index indicates an increase of blocking frequency






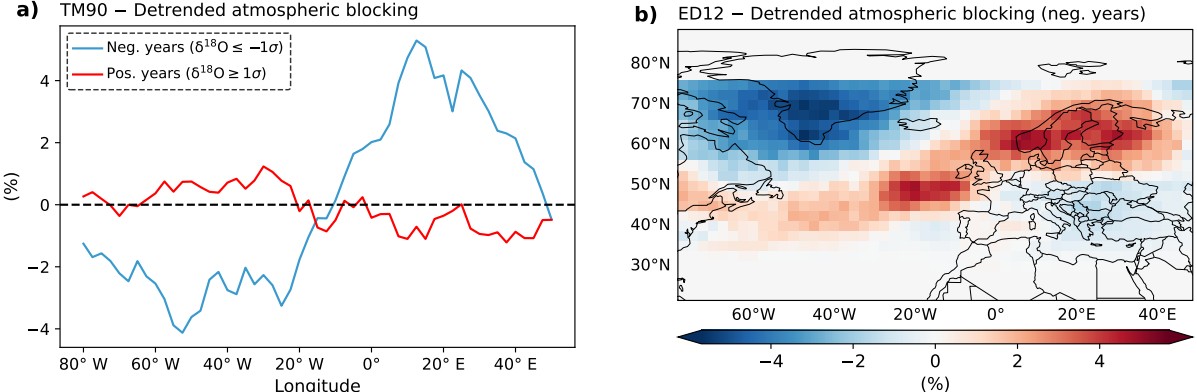

**Figure 3.** Detrended monthly averaged Tibaldi and Molteni (1990) **(a)** and Davini et al. (2012) **(b)** atmospheric blocking indices on the winter season (DJF) in the negative years the NGT stacked ice core record for the period $1920 - 2011$.

at around $60°$ N and between $10°$ W and $35°$ E, with its peak around $15° - 25°$ E. The ED12 index (Fig. 3b) permits also to analyze the spatial pattern of the atmospheric blocking. It corroborates an increase of blocking around the same region, the

Baltic Sea.

Given the clear atmospheric circulation pattern observed during the negative years of the NGT stacked $\delta^{18}O$ series, the blocking pattern highlighted by the two atmospheric blocking indices and the role of atmospheric blocking in favoring extreme weather events (Rex, 1950), the following analyses of temperature and precipitation effects will focus exclusively on negative years.

The increase of frequency of atmospheric blocking over Europe entails a change in the regional pattern of temperature and precipitation (Fig. 4). The position and spatial extent of atmospheric blocking facilitate the poleward transport of warm, moist air from the Atlantic Ocean (Fig. 4a-b) toward the Norwegian coast, leading to higher temperatures (Fig. 4c) and increased precipitation (Fig. 4d). The temperature increase results from warm-air advection, while the precipitation increase is driven by heat loss and the orographic uplift of these relatively warm air masses, which boosts condensation and precipitation formation.

The subsequent warming in Sweden (Fig. 4c) is attributed to adiabatic compression of the now dry air masses, a process associated with föhn winds.

Concurrently, the atmospheric blocking also facilitate advection of cold air masses from the Nordic region toward mid-latitudes, which leads to colder and drier conditions over southern Europe (Fig. 4c-d). As a result, temperatures drop in western Turkey, Greece, and, to a lesser extent, in southern Italy and the Iberian Peninsula. Additionally, moist air masses from the

Atlantic, which would typically influence southern Europe, are diverted toward Scandinavia, reducing precipitation in this region. Furthermore, central Europe shows a pattern of increased temperature and precipitation. This is likely due to a portion of Atlantic air masses reaching the region, contributing to both warming and wetter conditions.

It is also worth noting that the spatial distribution of TN10p over Scandinavia appears irregular. This irregularity is likely a result of the sparse coverage of the E-OBS dataset before 1950, which affects the computation of temperature indices and



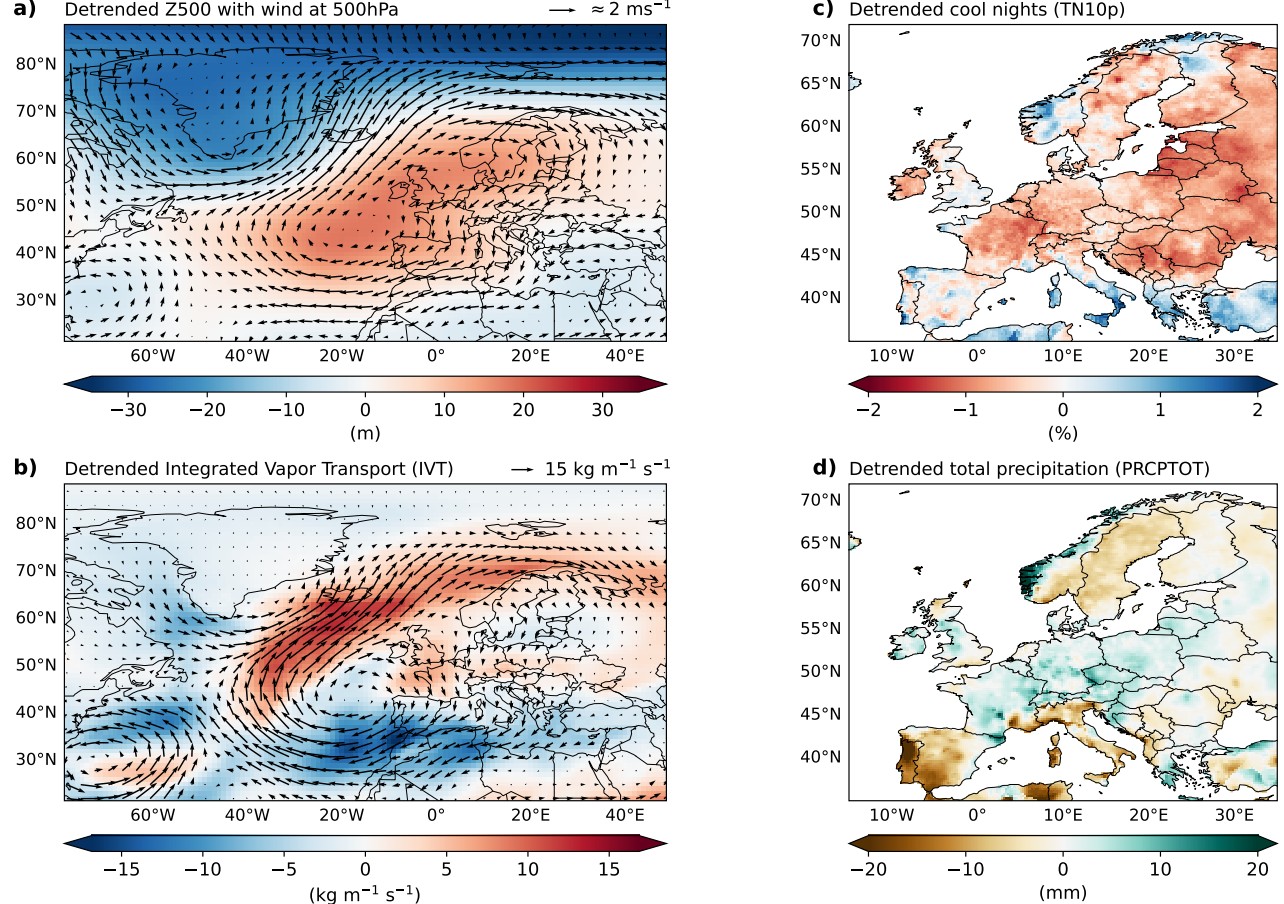

**Figure 4.** The composite maps of the detrended monthly anomalies of the geopotential height (shaded) at 500 hPa (Z500) and the wind (vector) at 500hPa **(a)**, detrended warm nights index TN10p **(b)**, detrended integrated vapor transport (IVT) **(c)**, and detrended total precipitation index PRCPTOT **(d)** on the winter season (DJF) in the negative years the NGT stacked ice core record for the period $1920 - 2011$.

leads to irregularities in the observed spatial patterns. On this purpose, the same analysis is carried out using CRAI dataset for TN10p, which shows consistent results with the results obtained using E-OBS dataset. Therefore, the sparsity of E-OBS before 1950 does not affect much the spatial patterns of temperature (Fig. A2).

## 3.2   Long-term period $(1602 - 2003)$

    The atmospheric circulation in the negative years during the long-perspective period (Fig. 5a) follows closely the atmospheric
circulation in the negative years of the observational period. The high-pressure system is present over Europe and in particular over the northern France and southern Great Britain. The particular circular shape of the high-pressure system has to be ascribed on average conditions as discussed in the previous section. In this case, the presence of the atmospheric blocking is not possible





**Figure 5.** The composite maps of the detrended monthly anomalies of the geopotential height (shaded) at 500 hPa (Z500) and the wind (vector) at 250hPa **(a)**, detrended 2-meter temperature **(b)**, and detrended total precipitation **(d)** on the winter season (DJF) in the negative years the NGT stacked ice core record for the period $1602 - 2003$.

to assess since the EKF dataset provides only monthly data, and therefore, the TM90 and ED12 indices cannot be computed. On the other hand, it is reasonable to associate such high pressure patterns with increased frequency of synoptic-scale blocking 170 circulation in the corresponding region.

The physical explanation outlined in the previous section for the observational period aligns well with the effect of the atmospheric blocking on temperature and precipitation over Europe observed in the long-term perspective period. In particular, it is observed a rise in temperature and increase in precipitation in Norwegian coastal areas due to atmospheric transport of moist and relative warmer air masses. On the contrary, in southern Europe a decrease in precipitation due redirection of 175 the moist air masses toward the Nordic regions and a decrease in temperature due to cold advection of Nordic air masses is





observed. The observed patterns in temperature and precipitation results to be more regular than the observational period due to the use, in this case, of a reanalysis product.

While the high-pressure system over Europe and the low-pressure system (see Fig. A3a) result to be statistically different from zero, the main features of the effect of the high-pressure system on temperature and precipitation are not. The consistency
of the results between the observational and long-term perspective periods and the statistical significance of the geopotential height pattern suggests a non-significant shift in the mean.

To further investigate whether the observed atmospheric blocking affects temperature and precipitation, the distribution of weighted averages of temperature and precipitation in the following selected grid boxes is estimated using the method of Kernel Density Estimation (KDE):

(i)  "central Europe" for the geopotential height ($38° - 55°$ N and $5°$ W $- 20°$ E);

(ii)  "Baltic region" for temperature ($50° - 64°$ N and $7° - 23°$ E);

(iii)  "Scandinavia" for precipitation ($60° - 70°$ N and $7° - 30°$ E);

(iv)  "southern Europe" for precipitation ($35° - 50°$ N and $10°$ W $- 50°$ E).

The distribution of the indices derived from these four boxes corroborate the hypothesis of a change in the distribution of
the temperature and precipitation (Fig. 6). In particular, the geopotential height index over the box "central Europe" is not only statistically different from zero, but also shows notable differences in comparison with the distribution of the positive years (Fig. 6a). The distribution of the temperature index over the box "Scandinavia" shows a slightly change toward warmer temperatures compared to the positive years, but not so evident (Fig. 6b). On the contrary, the distribution of the precipitation index over the box "Scandinavia" clearly exhibits a change toward more heavy rainfall conditions, while the distribution of the
precipitation index over the box "southern Europe" shows a remarkable change toward drier conditions compared to positive years. The changes in the distribution in the precipitation index in the box "Scandinavia" and "Souther Europe" results to be statistically significant from their corresponding distribution during positive years.

Since EKF dataset provides monthly data, it is possible to investigate further if the change in distribution of precipitation occurs among all winter months or only throughout some particular months. And possibly, if the lack of change in distribution
of temperatures averages out among the months.

The probability density function of the indexes is then estimated for the indexes computed from the boxes "central Europe", "Baltic region", "Scandinavia" and "southern Europe", as defined above (Fig. 7). The distribution of the geopotential height exhibits a change toward positive values, meaning more frequent high-pressure system in the box "central Europe" mainly in January and February (Felis et al., 2018). Consistently, the distributions of the precipitation over the box "Scandinavian"
and over the box "southern Europe" remarkably change toward wetter and drier conditions, respectively. On the contrary, the temperature index over the box "Baltic region" shows a slight change towards warmer temperatures.





**Figure 6.** Probability density functions of the standardized monthly field average for the following atmospheric variables: geopotential height at 500 hPa (Z500) **(a)**, 2-meter temperature over the selected box **(b)**, and precipitation **(c, d)**). All indices are calculated for winter (DJF) during the negative years of the NGT-stacked ice core $\delta^{18}O$ record, covering the period 1602–2003. The shaded bands represent the 90% confidence intervals. The corresponding analysis boxes are shown with solid red lines in the upper-right map of each panel. See the main text for the coordinates used for the NGT-stacked ice record.



**Figure 7.** Comparison of the estimated probability density functions of the monthly standardized field average **(a-c)** of the atmospheric variables depicted in Fig. 6 **(1-4)** in DJF in the negative years the NGT stacked ice core record for the period $1602 - 2003$. Shaded areas are the 90% confidence interval.





## 4 Conclusions

The large-scale atmospheric circulation associated with the extremely high and low values observed in the NGT stacked $\delta^{18}$O series during the observational $(1920 - 2011)$ and long-term $(1602 - 2003)$ periods is investigated using composite analysis. The atmospheric circulation during the negative years for the observational period shows an anomaly pattern resembling an atmospheric blocking over Europe extended from Azores Islands to Baltic Sea.

The absence of a consistent large-scale atmospheric circulation pattern during years with positive $\delta^{18}$O anomalies highlights that low $\delta^{18}$O values in the NGT stacked $\delta^{18}$O time series are more strongly influenced by specific atmospheric circulation features, such as atmospheric blocking. It is worth noting that the circulation pattern shown by the geopotential height (Fig. 2b) may resemble a positive phase of the NAO. However, the positive phase of the NAO usually is characterized by a widespread positive anomaly in geopotential height, rather than a more localized high-pressure system centered over Europe. The relationship between the NAO and variability in ice cores has been studied in depth (e.g., Appenzeller et al., 1998; Vinther et al., 2003a). In this study, however, the ice core data are based on a stacked record, mainly composed of ice cores from central and Northern Greenland. The specific locations of these ice cores, along with the stacking method, which helps to reduce the influence of local processes on the variability of $\delta^{18}$O, allow for a better representation of different atmospheric circulation patterns, rather than just the NAO.

The atmospheric blocking indices TB90 and ED12 corroborate that the anomaly in the geopotential height are associated with an increase in atmospheric blocking over the Baltic Sea. The atmospheric blocking anomaly pattern is found to have an impact on the hydroclimatic conditions over Europe in the observational period. In particular, in the observational period, the results exhibit an average increase of precipitation in Norwegian coastal and drier conditions over southern Europe. This provides good empirical evidence supporting that NGT stacked $\delta^{18}$O is a good proxy for hydroclimatic conditions over Europe (Pfahl, 2014). However, the limited amount of observations does not allow to carry out statistical significance.

The availability of paleoclimate reanalysis permits to assess whether the pattern of the atmospheric blocking over Europe present in the observational period is also present in the long-term perspective period $(1602 - 2003)$. Results in Sect. 3.2 outline an analogous pattern of an high-pressure system centered over northern France and Southern Great Britain that increases precipitation in the Norwegian coastal areas and decreases precipitation in southern Europe. This high-pressure system pattern result to be statistical significant (Fig. A3). Distributions of box-based indices for geopotential height, temperature, and precipitation in the areas reveal a statistical significant change in the hydroclimatic conditions. In particular, wetter conditions for Norwegian coastal areas and drier conditions for southern Europe are observed. Wetter and drier conditions are associated with warmer and colder conditions. Moreover, the period of major change in the distribution of the precipitation during the winter is January.

This study demonstrates that the NGT-stacked $\delta^{18}$O time series reliably captures signals of extreme hydroclimatic conditions in Europe, mainly through its variability linked to atmospheric blocking patterns. The consistent association between low $\delta^{18}$O anomalies and blocking-induced precipitation changes across both observational and long-term periods provides robust evidence for using $\delta^{18}$O as a proxy to reconstruct past European climate variability and extremes. A logical next step would



be to extend this approach to other ice core records and integrate high-resolution climate models to better understand regional impacts of blocking events under future warming scenarios.

*Code availability.* The code of the analysis is available contacting the corresponding author upon request.

*Data availability.* The 20th Century Reanalysis (V3) (Slivinski et al., 2019) is freely available online(https://psl.noaa.gov/data/gridded/
245    data.20thC_ReanV3.html). The CRAI dataset (Plésiat et al., 2024) is freely available in the link indicated in the publication. The 1D and 2D atmospheric blocking algorithm is available only (https://github.com/oloapinivad/MiLES). The paleo-reanalysis dataset EKF400v2 is available at Franke et al. (2020). The E-OBS (Cornes et al., 2018) data is freely available online https://surfobs.climate.copernicus.eu/ dataaccess/access_eobs.php. The Python package icclim (Aoun et al., 2024) is freely available online (https://github.com/cerfacs-globc/ icclim).





250 **Appendix A: Supplementary information**

**Figure A1.** As Fig. 2, but panel **(c)** and **(d)** are, respectively, the detrended minimum temperature and total precipitation using 20CRv3.



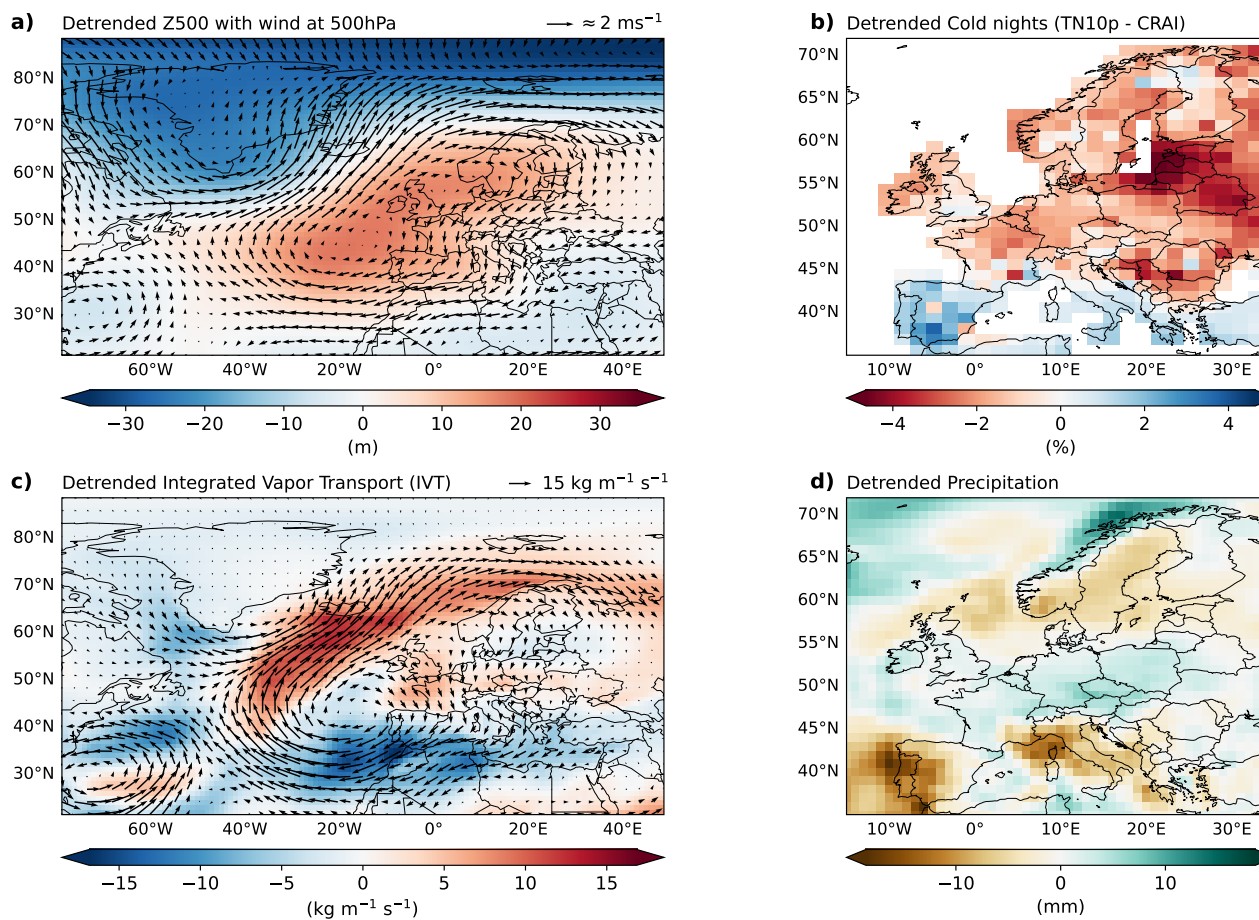

**Figure A2.** As Fig. A1, but panel **(b)** uses CRAI dataset





**Figure A3.** The composite maps of the detrended monthly anomalies of the geopotential height (shaded) at 500 hPa (Z500) and the wind (vector) at 200hPa on the winter season (DJF) in the negative years the NGT stacked ice core record for the period: **(a)** $1602 - 2003$, **(b)** $1601 - 1700$, **(c)** $1701 - 1800$, **(d)** $1801 - 1900$, and **(e)** $1901 - 2003$. Hatched areas in panel **(a)** are statistically different from zero at $90\%$ also taking into account the false discovery rate (FDR) at $10\%$ (Wilks, 2016).



*Author contributions.* Conceptualization and methodology, A.G., N.R, M.I.; Formal analysis, A.G., N.R, M.I.; Investigation, A.G., N.R, M.I.; Resources, A.G., N.R, M.I., G.L.; Data curation, A.G.; Writing–original draft preparation, A.G.; writing–review and editing, A.G, N.R., G.L., M.I.; All authors have significantly contributed to the preparation of this manuscript.

*Competing interests.* The authors declare no competing interests.

255  *Acknowledgements.* A.G. was supported by the International Science Program for Integrative Research in Earth Systems (INSPIRES) at AWI, via the project "Greenland Ice Cores as Proxy for European Extremes". M.I., G.L. and N.R. were partially supported by Helmholtz Association through the joint program "Changing Earth – Sustaining our Future" (PoF IV) program of the AWI. M.I. was also supported by the Helmholtz Climate Initiative REKLIM. The authors gratefully acknowledge the German Climate Computing Centre (DKRZ) for providing computing time on the supercomputer Levante.



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
