# Peer review of "Northern Greenland transect stacked ice cores as a proxy for winter extreme events in Europe"

_EGUsphere, 2025_

## Referee Comment (RC1)

**Review of Northern Greenland transect stacked ice cores as a proxy for winter extreme events in Europe**

In this manuscript, Gagliardi et al combine a stack of ice core records and a paleoclimate reanalysis to study past regimes of atmospheric circulation, and in particular conditions of atmospheric blocking suitable to general extreme events in Europe. The study uses both datasets from observational period (1920 to present) and from a *long term perspective* (1602 to 2011) and shows that the atmospheric circulation patterns over the two periods are relatively similar.

The manuscript is well written, the analyses are sound and appropriate to study the dynamical systems. From a paleoclimate point of view, I feel that the manuscript is barely scratching the surface and that the manuscript as it is now is missing a discussion: the results are clear, but their consequences is not discussed: what does it mean that similar atmospheric blocking conditions can be found in both the 1602-2011 and the observational period? Considering that $\delta^{18}O$ is also a temperature proxy (Hörhold et al., 2023), is it possible to disentangle the blocking conditions (characterised with $\delta^{18}O$ below -1$\sigma$) happening less/more often with colder/warmer conditions from the thermodynamical response of water isotopes? i.e. with a warmer average conditions, should the -1$\sigma$ threshold also evolve? And finally, and maybe the more importantly, can you conclude anything on the impact of the anthropogenic climate change on the frequency of extreme events in Europe from the stack and the reconstruction?

I believe that this manuscript would be a great addition to Climate of the Past once these questions are answered. I include general and specific comments below.

**General comments:**

As discussed above, I feel like the manuscript is missing a discussion that would put the results into a larger context, as well as to discuss the limitations of the datasets used. In particular, I believe that answering the following questions would be beneficial to the study:

1. The long term period analysis is based on a paleoreanalysis, which rely on a dataset of proxy records and instrumental measurements. Before 1850, the reanalysis is constructed using almost solely tree ring records. How do the limitations of using tree rings for the paleoreanalysis affect your results? Typically, in order to reconstruct fields of atmospheric circulation patterns, temperature, and precipitation, the EFK v2 makes use of ECHAM5.4 and these datasets, but this is still a reanalysis based on a limited set of proxies which themselves have some well-known biases in term of reconstructing variability, including the change of growth rate for the different life stage of the trees leading to non-linearity in the relationship between the tree ring growth and isotopic composition and the local climatic conditions.

2. It also raises the question of the weight of the reconstruction from tree rings in the paleoreanalaysis. The consistency between the results from the observational period and the long term perspective periods (lines 179-180) could also be linked with the tree ring reconstruction might share some of the variance of the ice core stack, are just representative the same mode of variability. While it's beyond the scope of the manuscript to compare the tree ring variability with the ice core variability, I think that a critical discussion of the impact of the tree ring reconstructions on the paleoreanalysis in the framework of comparing it with another paleoclimate reconstruction could be valuable.

3. The NAO is mentioned once in the introduction and then not a single time in the manuscript before the conclusion where an entire paragraph discusses the link between the atmospheric patterns described here and the NAO. The conclusion should not include new information, and this highlight the lack of discussion section in the manuscript. In term of content, how does the NGT stacked $\delta^{18}O$ compare with NAO indices (Ortega et al., 2015)? The paragraph in the conclusion doesn't appear convincing: all the ice cores from the stack should be under the same influence of NAO patterns considering the relatively small area in which they were found (Casado et al., 2013).

4. Since you have a 400-year reconstructions, how are the blocking conditions changing over time? Is there a link between the temperature (which also affects the NGT stacked $\delta^{18}O$) and the blocking conditions?

5. (Hörhold et al., 2023) identified a large warming in Greenland, with an increase of $\delta^{18}O$. Here, you are using detrended $\delta^{18}O$, which remove this effect. Nonetheless, it should be discussed that the minima, in particular the recent ones in the 1980s are actually associated with values close to 0‰. In particular, two

aspects are key to be mentioned: (i) (Hörhold et al., 2023) shows that there is a regime change with a trend changing around 1800, so the detrending from 1602 – 2011 is not necessarily physically based, how does the window used for the reference trend is affecting your results? and (ii) how does a warmer baseline affect your results? Overall, it's not clear to me after reading the article if the detrended NGT variability is a direct signal from the atmospheric circulation, or temperature variability in Greenland that happens to be, at least partly, correlated to atmospheric circulation.

**Specific comments:**

Lines 28 to 29: "When such extreme events persist over a region for extended periods, they can be classified as extreme climate events."

While I feel the goal here is to distinguish between extreme weather events and extreme climate events, the sentence is not very clear.

Line 31: "However, the lack of high temporal resolution in proxies data makes a challenge reconstructing weather extreme events."

I don't think proxies data is a clear concept. Paleoclimate records maybe?

Lines 31 to 35: Overall, this paragrap seems a bit weaker than the rest of the introduction, because it seems you're not saying what you want. Since it's not clear what you are studying here, it's not clear to see why tree ring reconstructions are limited. In the abstract, you mention 1602 to 2011, but there are multiple reconstructions from tree ring covering this time span, for instance, Freund et al, 2023 covers exactly this window.

Line 36: "Ice cores records can be used for multidecadal and longer time scale reconstructions (Rimbu and Lohmann, 2010b)."

Yes, but this is not maybe the most relevant citations for this, and seems to promote self citation quite a lot. Clearly, papers ranging from Vinther et al, 2010 to GRIP/NEEM papers would be more relevant here.

Lines 36 to 38 "The growing number of high temporal resolution ice cores from the Greenland ice sheet gives valuable information on climate variations from seasonal to multidecadal time scales."

This sentence should be justified, but it's not clear to me that obtaining high resolution ice cores is new.

Lines 40 to 42: "Recent studies, though, have identified strong links between Greenland δ18O variability and atmospheric weather regimes (Rimbu and Lohmann, 2010a; Ortega et al., 2014) and relationship with atmospheric blocking during boreal winter months (Rimbu et al., 2007, 2017, 2021)."

10 to 15 years old studies cannot be really that recent. Overall, the introduction does not need to emphasize so much on how recent records are, but should focus on giving readers information about the important aspects of what can and cannot be done with ice cores.

Lines 44 to 45: "To this end, this paper assesses the validity of the δ18O variability in the Northern Greenland Transect (NGT) stacked ice cores (Hörhold et al., 2023) is a proxy for extreme climate events."

You mention reconstruction from 1600's to 2020's, while the NGT stack goes all the way back to 1000 AD. Why are you stopping there? It seems peculiar that you put so much value on the NGT stack, and not so much on the EKF paleoreanalysis which is as important if not more important to your analysis than the NGT stack.

Lines 128 to 130: "The average pattern in negative years features a high-pressure system extending from the Azores Islands to the Baltic Sea and low-pressure system over Greenland, whereas the average circulation pattern in positive years is not close to be the opposite of that in negative years."

It's difficult not to think of the link with NAO here.

Figure 2: Shouldn't there be a figure, at least in supplement that show the reference against which the anomalies have been plotted ? Here, it's difficult to know for instance if the changes are equivalent to less strong winds toward Europe in negative years, or actually an opposite wind direction.

Lines 141 to 144: "Given the clear atmospheric circulation pattern observed during the negative years of the NGT stacked $\delta18O$ series, the blocking pattern highlighted by the two atmospheric blocking indices and the role of atmospheric blocking in favoring extreme weather events (Rex, 1950), the following analyses of temperature and precipitation effects will focus exclusively on negative years."

I'm not sure that this is a very sound argument, yes it peaks around 5% for the negative years in Fig 3a versus 1% for the positive years, but 1% is still quite a large number of occurence. Since it's over the ocean mostly, the effects aren't crucial, and you are more interested about Europe?

Lines 176 – 177: "The observed patterns in temperature and precipitation results to be more regular than the observational period due to the use, in this case, of a reanalysis product."

This sentence is unclear, are you talking about the long term perspective or something else, and also because the datasets used are reanalysis for both the observational period (20thcentury reanalaysis) and long term perspective (EFK v2 paleoreanalaysis).

---

## Referee Comment (RC2)

**Review of "Northern Greenland transect stacked ice cores as a proxy for winter extreme events in Europe", by Gagliardi et al.**

This short paper discusses how a Greenland isotopic data can record information on winter blocking events over the North Atlantic region. The paper is based on isotopic data and a long reanalysis, and performs various statistical analyses. The authors propose physical interpretations by determining how water is transported in the atmosphere.

The paper is interesting and fits nicely in the scope of Climate of the Past. I have a few remarks that could be integrated easily.

**Major comments**

Investigating the relation between this isotopic record and blocking events and the consequences on surface variables is probably innovative. The authors mention very recent references, which is fine, but could also have searched for references at the turn of the 21$^{st}$ century, who looked at relations between the atmospheric circulation and surface variables, e.g. (Meeker et al., 1997). The relationship between atmospheric patterns and surface extremes has been investigated since (Robertson and Ghil, 1999; Yiou et al., 2012; Yiou and Nogaj, 2004), just to cite a few. And the relation between the jet stream and European extremes was recently discussed by (Xu et al., 2024). Therefore, a more thorough bibliographic search would certainly be welcome, to put the results of the paper in a fair perspective.

The adjective "extreme" appears in the title and in several instances of the manuscript. The only extremes that are discussed are the values of the isotopic record, not hydrological or temperature extremes in Europe. The authors essentially discuss "warmer/colder" or "wetter/drier" than normal, which does not correspond to usual definitions of extremes. This should be amended in the manuscript.

The authors quickly deduce from Figure 2 that the relation between the isotopic record and the atmospheric circulation is unequivocal. In order to draw any conclusion between local (European) surface variability and blockings in the past, the authors should also determine the expected value of the isotopic record conditional on the occurrence of blocking (what they compute is Z500 conditional on the value of the isotopic record).

The paper could also have discussed a few key events that occurred since 1600, including volcanic eruptions, solar minima, etc.

**Specific comments**

l. 70: the description of PRCPTOT is not clear. Cumulated over what time scale?

l. 73: the data description is not very informative. What is the input of the AI reconstruction? What is its added value here?

l. 84: I do not understand what "[...] series assumes high and low values according to certain thresholds." Please rephrase.

l. 86: Why does a +/- 1 sigma threshold meet "both criteria"?

Figure 1 (and text): how is sigma computed? What period? sigma obviously increases with time in Fig. 1c.

Eqs. (1) and (2): I assume that the blocking indices are determined on daily time scales. Most papers (including (Tibaldi and Molteni, 1990)) use a lowercase \phi for latitude.

l. 126: Here, and in many other places, the authors are very qualitative: Figure 2a shows a cyclonic anomaly over the North Atlantic (albeit not as deep as the cyclonic anomaly over Greenland in Fig. 2b). The absence of symmetry in the maps of Fig. 2 is not very surprising. The values of Z500 and wind speed anomalies are symmetric over Greenland, though, which is the first criterion expressed by anomalies of the isotopic record. Since the North Atlantic atmospheric circulation goes eastward, and yields geostrophic features (regardless of the presence of a blocking feature), no real symmetry of the Z500 field east of Greenland should be expected.

l. 170: The association between high pressure patterns with increased frequency of synoptic-scale blocking circulation is demonstrated by (Yiou and Nogaj, 2004).

l. 178—206: the discussion is very qualitative, with many adverbs ("clearly", "remarkably", "notably", etc.) that could be assorted with numbers, to reach objectiveness.

l. 199: verb missing in sentence.

Figures 6 and 7, l. 237: the results that are reported do not say anything about extremes, which are in the tails of the distributions. None of the figures show any change in the tails of distributions. It is already interesting to discuss how the centers of the distributions change.

l. 240: why would it be "logical" to extend this study to other ice cores? Would any change (especially for other Greenland ice cores) be expected? If so, this would rather invalidate the whole approach, wouldn't it? As a perspective, what would seem natural (to me), would be to investigate the how natural forcings can affect features of the atmospheric circulation. This issue is barely discussed in the manuscript, while it is a key aspect of paleoclimate studies.

**References**

Meeker, L. D., Mayewski, P. A., Twickler, M. S., Whitlow, S. I., and Meese, D.: A 110,000-year history of change in continental biogenic emissions and related atmospheric circulation inferred from the Greenland Ice Sheet Project Ice Core, J. Geophys. Res. Oceans, 102, 26489–26504, 1997.

Robertson, A. and Ghil, M.: Large-scale weather regimes and local climate over the western United States, J Clim, 12, 1796–1813, 1999.

Tibaldi, S. and Molteni, F.: On the operational predictability of blocking, Tellus A, 42, 343–365, 1990.

Xu, G., Broadman, E., Dorado-Liñán, I., Klippel, L., Meko, M., Büntgen, U., De Mil, T., Esper, J., Gunnarson, B., Hartl, C., Krusic, P. J., Linderholm, H. W., Ljungqvist, F. C., Ludlow, F., Panayotov, M., Seim, A., Wilson, R., Zamora-Reyes, D., and Trouet, V.: Jet stream controls on European climate and agriculture since 1300 ce, Nature, 634, 600–608, https://doi.org/10.1038/s41586-024-07985-x, 2024.

Yiou, P. and Nogaj, M.: Extreme climatic events and weather regimes over the North Atlantic: When and where?, Geophys Res Lett, 31, L07202, https://doi.org/10.1029/2003GL019119, 2004.

Yiou, P., Garcia de Cortazar-Atauri, I., Chuine, I., Daux, V., Garnier, E., Viovy, N., van Leeuwen, C., Parker, A. K., and Boursiquot, J. M.: Continental atmospheric circulation over Europe during the Little

Ice Age inferred from grape harvest dates, Clim Past, 8, 577–588,  https://doi.org/DOI 10.5194/cp-8-577-2012,  2012.

---

## Author Comment (AC1)

**General comments**

I read this interesting and well-written study with great interest. I would like to use the opportunity of open community comment to ask for clarification of certain less clear points. Some of these questions are resonating with some points raised already by Mathieu Casado but proposed from a different perspective.

So, links between firn and ice core stable isotope time series from Greenland and weather/climate variability over Europe have been intensively studied during the past decades. Several excellent results were delivered just right from the team of the authors of the current paper, as well.

In this study a relatively new stacked  $\delta^{18}O$  record is tested as a proxy European hydroclimatic extremes driven by atmospheric blocking. The firn and ice cores used to compile the stacked  $\delta^{18}O$  record were actually distributed across north and central Greenland (Hörhold et al., 2023). The spatial extent of the ice core array scratches from ~71 to 80N and from ~30 to 50 W. However, different regions of the Greenland Ice Sheet experience distinct changes in the average moisture source locations, which is manifested in the stable isotopic signal of ice/firn (Sodemann et al., 2008 https://doi.org/10.1029/2007JD008503 ). Different regions of the Greenland Ice Sheet experience distinct changes in average moisture source locations, with the strongest variability observed in north and west Greenland. This strong sensitivity of Greenland winter precipitation to the NAO has a substantial impact on the stable isotope composition of Greenland precipitation, which is crucial to consider when interpreting ice-core  $\delta^{18}O$  records.

These isotope-hydrometeorological domains reflect the major regions of the Greenland delineated based on a synoptic survey of precipitation and possible effects of orography on moving cyclones (Chen et al., 1997 https://doi.org/10.1175/1520-0442(1997)010<0839:POGRBA>2.0.CO;2).

However, the employed stacked  $\delta^{18}O$  record amalgamates distinct regions considering the isotope-NAO relationship. My question is whether a subset of the firn/ice core  $\delta^{18}O$  records of Hörhold et al., 2023, rather than the NGT-2012 composite, might provide a better proxy for the European hydroclimatic extremes driven by atmospheric blocking?

Supporting the regionalization approach a previous study (Ortega et al., 2014, which is already cited in this manuscript) evaluated a set of shallow ice core derived  $\delta^{18}O$  series from Greenland also pointed out spatially variable depletion/enrichment in the isotopic composition of ice/firn depending on the dominant NAO mode. In addition, ice core d18O records across Greenland exhibited distinctive spatial arrangements regarding change points, all corresponded to changepoints in the NAO, indicative of a consistent atmospheric influence over the past millennium (Hatvani et al., 2022 https://doi.org/10.3390/atmos13010093 ).

We would like to thank the reviewer for the appreciation/suggestions/comments/feedback that will help us improve our manuscript, and for taking the time to read and review our paper.

The interesting point of view of stacking a subset of ice cores used for NGT-2012 is unfortunately beyond the scope of the paper. However, we agree with the comment that a ad-hoc selection of ice cores may be a better proxy for atmospheric blocking events over Europe.

**Specific comments**

The "Results" section actually contains some discussion so section title could be changed. In addition, to improve the discussion the results, beside the previously mentioned studies, a very recent paper (Brönnimann, et al. (2025). https://doi.org/10.1038/s41561-025-01654-y) largely overlapping with the current one could be compared.

**We are going to discuss the recent paper as suggested.**

The "main set of extremely consecutive positive years between 1927 and 1932" mentioned in line 115 corresponds well with the changepoint identified around 1933 in ice-core  $\delta^{18}$ O records from southern and central Greenland (Hatvani et al., 2022).

**We thank the reviwer and we will include in this in the discussion.**

The last comment on this part is that the text in lines 182-188 that could fit better to Methods. In addition, region names and citations of Fig 6 panels might need careful checking in the subsequent paragraphs (e.g. "Scandinavia" should be replaced with "Baltic" in line 192 however then

**We are going to modify the text as suggested.**

Minor technical issues with the reference list are:

- Authors' name for the !rst reference (line261) should be checked
- One of the duplicated items (lines 321-325; lines 349-352) should be removed from the reference list

In the revised version of the manuscript we are going to correct the aforementioned technical issues.

---

## Author Comment (AC2)

**Review of Northern Greenland transect stacked ice cores as a proxy for winter extreme events in Europe**

In this manuscript, Gagliardi et al combine a stack of ice core records and a paleoclimate reanalysis to study past regimes of atmospheric circulation, and in particular conditions of atmospheric blocking suitable to general extreme events in Europe. The study uses both datasets from observational period (1920 to present) and from a long term perspective (1602 to 2011) and shows that the atmospheric circulation patterns over the two periods are relatively similar.

The manuscript is well written, the analyses are sound and appropriate to study the dynamical systems. From a paleoclimate point of view, I feel that the manuscript is barely scratching the surface and that the manuscript as it is now is missing a discussion: the results are clear, but their consequences is not discussed: what does it mean that similar atmospheric blocking conditions can be found in both the 1602-2011 and the observational period? Considering that  $\delta^{18}$ O is also a temperature proxy (Hörhold et al., 2023), is it possible to disentangle the blocking conditions (characterised with  $\delta^{18}$ O below -1 $\sigma$ ) happening less/more often with colder/warmer conditions from the thermodynamical response of water isotopes? i.e. with a warmer average conditions, should the -1 $\sigma$  threshold also evolve? And finally, and maybe the more importantly, can you conclude anything on the impact of the anthropogenic climate change on the frequency of extreme events in Europe from the stack and the reconstruction? I believe that this manuscript would be a great addition to Climate of the Past once these questions are answered. I include general and specific comments below.

We would like to thank the reviewer for the appreciation/suggestions/comments/feedback that will help us improve our manuscript, and for taking the time to read and review our paper.

**General comments**

As discussed above, I feel like the manuscript is missing a discussion that would put the results into a larger context, as well as to discuss the limitations of the datasets used. In particular, I believe that answering the following questions would be beneficial to the study:

1. The long term period analysis is based on a paleoreanalysis, which rely on a dataset of proxy records and instrumental measurements. Before 1850, the reanalysis is constructed using almost solely tree ring records. How do the limitations of using tree rings for the paleoreanalysis affect your results? Typically, in order to reconstruct fields of atmospheric circulation patterns, temperature, and precipitation, the EFK v2 makes use of ECHAM5.4 and these datasets, but this is still a reanalysis based on a limited set of proxies which themselves have some well-known biases in term of reconstructing variability, including the change of growth rate for the different life stage of the trees leading to non-linearity in the relationship between the tree ring growth and isotopic composition and the local climatic conditions.

The EKF v2 reconstruction is indeed mainly based on tree rings before 1850. However,

the average geopotential height pattern during negative years in the NGT stack is consistent with the pattern observed in the instrumental period, where the reanalysis does not rely on proxies. In addition, over Greenland, the negative anomaly in geopotential height is also captured, despite the absence of tree-ring data in this region. It is also important to note that tree rings are generally better proxies for summer-based reconstructions. Therefore, the biases introduced by tree-ring proxies are mostly evident in the summer season. Moreover, the EKF re-analysis assimilates also other types of proxies (e.g., corals) as well as old documentary evidence.

2. It also raises the question of the weight of the reconstruction from tree rings in the paleoreanalysis. The consistency between the results from the observational period and the long term perspective periods (lines 179-180) could also be linked with the tree ring reconstruction might share some of the variance of the ice core stack, are just representative the same mode of variability. While it's beyond the scope of the manuscript to compare the tree ring variability with the ice core variability, I think that a critical discussion of the impact of the tree ring reconstructions on the paleoreanalysis in the framework of comparing it with another paleoclimate reconstruction could be valuable.

As stated by the reviewer, the interesting question on how tree ring reconstructions impact paleoreanalysis is beyond the scope of the paper. However, we are going to try to make it clearer in the manuscript about the limitations of the tree ring in the paleoreanalysis.

3. The NAO is mentioned once in the introduction and then not a single time in the manuscript before the conclusion where an entire paragraph discusses the link between the atmospheric patterns described here and the NAO. The conclusion should not include new information, and this highlight the lack of discussion section in the manuscript. In term of content, how does the NGT stacked  $\delta$ 18O compare with NAO indices (Ortega et al., 2015)? The paragraph in the conclusion doesn't appear convincing: all the ice cores from the stack should be under the same influence of NAO patterns considering the relatively small area in which they were found (Casado et al., 2013).

The plot below shows the 31-year rolling cross-correlation between the NAO reconstruction by Ortega et al. (2025) and the  $\delta^{18}$ O records from the NGT stack and DYE3 ice cores (Rasmussen et al., 2022). Negative years are marked as dots, which are plotted over the extreme negative years (-1 $\sigma$ ) in the NGT stack. Overall, the cross-correlation between the reconstructed NAO index and the NGT stack is, for most periods, lower than that between the reconstructed NAO and the DYE3 core. The reviewer's comment allows us to rephrase the manuscript to clarify that the NGT stack is not completely unaffected by the NAO. Our point is that the stacked signal from northern Greenland ice cores reflects not only the NAO but also the increase in atmospheric blocking events occurring over Europe. In the revised version of the manuscript we will add more information about NAO in relationship with the NGT timeseries.

4. Since you have a 400-year reconstructions, how are the blocking conditions changing over time? Is there a link between the temperature (which also affects the NGT stacked  $\delta$ 180) and the blocking conditions?

The atmospheric blocking events appear to be quite stable, as shown in Supplementary Figure A.3. The only period where there is less agreement with other intervals is the transition phase following the end of the Little Ice Age (LIA).

The connection between temperature and blocking events in Europe arises from the fact that, during such events, less moisture and relatively warmer air masses from the mid-latitudes are able to reach Greenland. Consequently, the relationship between temperature in northern Greenland and temperature in Europe is also mediated through atmospheric blocking events. We will also add this information in the revised version of the manuscript.

- 5. (Hörhold et al., 2023) identified a large warming in Greenland, with an increase of  $\delta$ 180. Here, you are using detrended  $\delta$ 180, which remove this effect. Nonetheless, it should be discussed that the minima, in particular the recent ones in the 1980s are actually associated with values close to 0‰. In particular, two aspects are key to be mentioned: (i) (Hörhold et al., 2023) shows that there is a regime change with a trend changing around 1800, so the detrending from 1602 2011 is not necessarily physically based, how does the window used for the reference trend is affecting your results? and (ii) how does a warmer baseline affect your results? Overall, it's not clear to me after reading the article if the detrended NGT variability is a direct signal from the atmospheric circulation, or temperature variability in Greenland that happens to be, at least partly, correlated to atmospheric circulation.
  - (i) We agree that a linear detrending may not fully capture the change around 1800. At the same time, we acknowledge a shift in the NGT stack during this period and tested some data-driven change detection methods. However, applying such approaches would introduce further subjectivity, since the outcome depends on the choice of method. In our tests, using a data-driven method for the period 1750–2011 (https://pypi.org/project/pwlf/), the detected change in trend occurs around 1991/1992. This mainly affects the last 20 years of the series, where only one year

shows an extremely low value. Therefore, we consider the effect of using a linear detrend on the results to be negligible.

(ii) We run again the composite analysis for not detrended NGT stack series. The results are essentially the same, therefore the baseline is not affecting much the dynamics.

**Specific comments**

| Lines 28 to 29: | When such extreme events persist over a region for extended periods, they can be classified as extreme climate events."                 |
|-----------------|-----------------------------------------------------------------------------------------------------------------------------------------|
|                 | While I feel the goal here is to distinguish between extreme weather events and extreme climate events, the sentence is not very clear. |
|                 | We are going to rephrase the sentence and make it clearer the concept between extreme weather and climate events.                       |
| Line 31:        | "However, the lack of high temporal resolution in proxies data makes a challenge reconstructing weather extreme events."                |

|                 | I don't think proxies data is a clear concept. Paleoclimate records maybe?                                                                                                                                                                                                                                                                                                                                                               |
|-----------------|------------------------------------------------------------------------------------------------------------------------------------------------------------------------------------------------------------------------------------------------------------------------------------------------------------------------------------------------------------------------------------------------------------------------------------------|
|                 | We are going to rephrase it.                                                                                                                                                                                                                                                                                                                                                                                                             |
| Lines 31 to 35: | Overall, this paragrap seems a bit weaker than the rest of the introduction, because it seems you're not saying what you want. Since it's not clear what you are studying here, it's not clear to see why tree ring reconstructions are limited. In the abstract, you mention 1602 to 2011, but there are multiple reconstructions from tree ring covering this time span, for instance, Freund et al., 2023 covers exactly this window. |
|                 | Tree rings are valuable proxies, but they mainly reflect summer conditions. In contrast, ice cores are more suitable for reconstructing winter climate, especially in the case of Greenland. We will rephrase the text to make this clearer.                                                                                                                                                                                             |
| Line 36:        | "Ice cores records can be used for multidecadal and longer time scale reconstructions (Rimbu and Lohmann, 2010b)."                                                                                                                                                                                                                                                                                                                       |
|                 | Yes, but this is not maybe the most relevant citations for this, and seems to promote self citation quite a lot. Clearly, papers ranging from Vinther et al, 2010 to GRIP/NEEM papers would be more relevant here.                                                                                                                                                                                                                       |
|                 | We are going to modify including more relavant citations.                                                                                                                                                                                                                                                                                                                                                                                |
| Lines 36 to 38: | "The growing number of high temporal resolution ice cores from the Greenland ice sheet gives valuable information on climate variations from seasonal to multidecadal time scales."                                                                                                                                                                                                                                                      |
|                 | This sentence should be justified, but it's not clear to me that obtaining high resolution ice cores is new.                                                                                                                                                                                                                                                                                                                             |
|                 | We are going to justify and rephrase the sentence.                                                                                                                                                                                                                                                                                                                                                                                       |
| Lines 40 to 42: | "Recent studies, though, have identified strong links between Greenland $\delta 180$ variability and atmospheric weather regimes (Rimbu and Lohmann, 2010a; Ortega et al., 2014) and relationship with atmospheric blocking during boreal winter months (Rimbu et al., 2007, 2017, 2021)."                                                                                                                                               |
|                 | 10 to 15 years old studies cannot be really that recent. Overall, the introduction does not need to emphasize so much on how recent records are, but should focus on giving readers information about the important aspects of what can and cannot be done with ice cores.                                                                                                                                                               |

|                   | We will include more details about the possibilities and limitations of using ice cores.                                                                                                                                                                                                                                                                  |
|-------------------|-----------------------------------------------------------------------------------------------------------------------------------------------------------------------------------------------------------------------------------------------------------------------------------------------------------------------------------------------------------|
| Lines 44 to 45:   | "To this end, this paper assesses the validity of the $\delta 180$ variability in the Northern Greenland Transect (NGT) stacked ice cores (Hörhold et al., 2023) is a proxy for extreme climate events."                                                                                                                                                  |
|                   | You mention reconstruction from 1600's to 2020's, while the NGT stack goes all the way back to 1000 AD. Why are you stopping there? It seems peculiar that you put so much value on the NGT stack, and not so much on the EKF paleoreanalysis which is as important if not more important to your analysis than the NGT stack.                            |
|                   | We limited the investigation period to 1600 due to the availability of the EKF paleoreanalysis dataset. However, we acknowledge the added value of discussing the NGT stack values before 1600, and we will include such a discussion.                                                                                                                    |
| Lines 128 to 130: | "The average pattern in negative years features a high-pressure system extending from the Azores Islands to the Baltic Sea and low-pressure system over Greenland, whereas the average circulation pattern in positive years is not close to be the opposite of that in negative years."                                                                  |
|                   | It's difficult not to think of the link with NAO here.                                                                                                                                                                                                                                                                                                    |
|                   | We agree that, as stated, it may resemble a purely NAO+ signal. However, our point is that the NGT stack reflects more than just NAO influence, unlike the southern Greenland ice cores. We will rephrase to clarify that our argument is not that NAO+ has little or no influence, but rather that the signal also reflects atmospheric blocking events. |
| Figure 2:         | Shouldn't there be a figure, at least in supplement that show the reference against which the anomalies have been plotted? Here, it's difficult to know for instance if the changes are equivalent to less strong winds toward Europe in negative years, or actually an opposite wind direction.                                                          |
|                   | We did not compute anomalies relative to a reference period because our goal was not to assess the impact of global warming. By applying a linear detrending over the entire periods, the observational period (1920–2011) and the long-term period (1602–2003), the fitted values (detrended series) naturally have a mean zero value.                   |
| Lines 141 to 144: | "Given the clear atmospheric circulation pattern observed during the negative years of the NGT stacked $\delta^{18}$ O series, the blocking pattern highlighted by the two atmospheric blocking indices and the role of atmospheric blocking in favoring extreme weather events (Rex, 1950),                                                              |

the following analyses of temperature and precipitation effects will focus exclusively on negative years."

I'm not sure that this is a very sound argument, yes it peaks around 5% for the negative years in Fig 3a versus 1% for the positive years, but 1% is still quite a large number of occurence. Since it's over the ocean mostly, the effects aren't crucial, and you are more interested about Europe?

We focus on the negative years in the geopotential pattern shown in Figure 2b, since these years display nearly four times more atmospheric blocking events compared to positive years. For this reason, we considered it more relevant to emphasize the negative years.

**Lines 176 to 177:**

"The observed patterns in temperature and precipitation results to be more regular than the observational period due to the use, in this case, of a reanalysis product."

This sentence is unclear, are you talking about the long term perspective or something else, and also because the datasets used are reanalysis for both the observational period (20thcentury reanalaysis) and long term perspective (EFK v2 paleoreanalaysis).

We agree on the fact that the sentece is not clear and we are going to rephrase it.

---

## Author Comment (AC3)

**General comments**

This short paper discusses how a Greenland isotopic data can record information on winter blocking events over the North Atlantic region. The paper is based on isotopic data and a long reanalysis, and performs various statistical analyses. The authors propose physical interpretations by determining how water is transported in the atmosphere.

The paper is interesting and fits nicely in the scope of Climate of the Past. I have a few remarks that could be integrated easily.

We would like to thank the reviewer for the appreciation/suggestions/comments/feedback that will help us improve our manuscript, and for taking the time to read and review our paper.

**Major comments**

Investigating the relation between this isotopic record and blocking events and the consequences on surface variables is probably innovative. The authors mention very recent references, which is fine, but could also have searched for references at the turn of the 21st century, who looked at relations between the atmospheric circulation and surface variables, e.g. (Meeker et al., 1997). The relationship between atmospheric patterns and surface extremes has been investigated since (Robertson and Ghil, 1999; Yiou et al., 2012; Yiou and Nogaj, 2004), just to cite a few. And the relation between the jet stream and European extremes was recently discussed by (Xu et al., 2024). Therefore, a more thorough bibliographic search would certainly be welcome, to put the results of the paper in a fair perspective.

We agree that additional literature should be cited in the draft. In the revised version of the manuscript, we will improve the introduction by including not only recent studies but also earlier works, and taking into account the reviewer's suggestions.

The adjective "extreme" appears in the title and in several instances of the manuscript. The only extremes that are discussed are the values of the isotopic record, not hydrological or temperature extremes in Europe. The authors essentially discuss "warmer/colder" or "wetter/drier" than normal, which does not correspond to usual definitions of extremes. This should be amended in the manuscript.

We agree that phrasing wamer or colder may indicate a change in the mean. However, using ETCCDI indexes imply already the occurrence of the extreme. Nonetheless, we are going to change the wetter/drier terms in more/less occurrence of extreme rainfall events periods.

The authors quickly deduce from Figure 2 that the relation between the isotopic record and the atmospheric circulation is unequivocal. In order to draw any conclusion between local (European) surface variability and blockings in the past, the authors should also determine the expected value of the isotopic record conditional on the occurrence of blocking (what they compute is Z500 conditional on the value of the isotopic record).

The occurrence of atmospheric blocking during low  $\delta^{18}O$  values is linked to a specific dipole pattern, with low pressure over Greenland and high pressure in a blocking configuration over Europe, which limits the transport of water vapor to Greenland. In contrast, other types of blocking over Europe may lead to different atmospheric circulation over the North Atlantic and therefore have less influence on  $\delta^{18}O$  variability. We propose computing the average blocking frequency over the same region where high pressure is observed in the instrumental period, and then averaging  $\delta^{18}O$  values in the NGT stack for those years when the blocking index is above or below  $1\sigma$ . The same approach can also be applied to the long-term reconstruction period.

The paper could also have discussed a few key events that occurred since 1600, including volcanic eruptions, solar minima, etc.

We will broaden the discussion by including known events.

**Specific comments**

I. 70: the description of PRCPTOT is not clear. Cumulated over what time scale?

PRCPTOT is defined as the cumulative amount of daily rainfall recorded on wet days over a given period. A wet day is defined as a day with more than 1 mm of rainfall. In this case, the period considered is each month of the winter season (December–February, DJF). For a better clarity, we will add this information in the data and Method section, in the revised version of the manuscript.

I. 73: the data description is not very informative. What is the input of the AI reconstruction? What is its added value here?

CRAI uses a U-shaped neural network composed of partial convolutions (see the referenced paper for construction details). The network was trained using historical simulations from Earth System Models (ESMs) in the CMIP6 archive. The main goal of the CRAI dataset is to reconstruct climate extreme indices, particularly for periods and regions with sparse observational data. We used this dataset to show that the same patterns observed in E-OBS also appear in CRAI, although with less homogeneity compared to the E-OBS. We will clarify in the text the added value of using CRAI.

I. 84: I do not understand what "[...] series assumes high and low values according to certain thresholds." Please rephrase.

We are going to modify the text as suggested.

I. 86: Why does a +/- 1 sigma threshold meet "both criteria"? Figure 1 (and text): how is sigma computed? What period? sigma obviously increases with time in Fig. 1c.

We tested several thresholds, namely 0.5, 0.75, and 1 sigma. The proportions of values exceeding these thresholds were approximately 31%, 23%, and 16%, respectively. Based on these results and the sample size during the observation period, we considered 1 sigma as the

level that represents extreme cases. We are going to report the definition of the corrected sample standard deviation for simplify the reading. As reported in L. 87, we compute the sample standard deviation over the whole observation and long-term perspective periods. Therefore, the extreme values in the NGT stack  $\delta^{18}$ O in the observation period are determined using the sample standard deviation over the observation period (1920 – 2011). This applies to the long-term perspective period (1602 – 2003).

Eqs. (1) and (2): I assume that the blocking indices are determined on daily time scales. Most papers (including (Tibaldi and Molteni, 1990)) use a lowercase \phi for latitude.

The atmospheric blocking index explained in the text refers to Davini et al. (2012). In their paper, the \phi latitude is in capital letter. For coherence in explaining such method, we adopted the same notation.

I. 126: Here, and in many other places, the authors are very qualitative: Figure 2a shows a cyclonic anomaly over the North Atlantic (albeit not as deep as the cyclonic anomaly over Greenland in Fig. 2b). The absence of symmetry in the maps of Fig. 2 is not very surprising. The values of Z500 and wind speed anomalies are symmetric over Greenland, though, which is the first criterion expressed by anomalies of the isotopic record. Since the North Atlantic atmospheric circulation goes eastward, and yields geostrophic features (regardless of the presence of a blocking feature), no real symmetry of the Z500 field east of Greenland should be expected.

Thank you for this comment. We are going to rephrase and improve our descriptions in the revised version of the manuscript.

I. 170: The association between high pressure patterns with increased frequency of synoptic-scale blocking circulation is demonstrated by (Yiou and Nogaj, 2004).

We thank the reviewer for this suggestion.

I. 178—206: the discussion is very qualitative, with many adverbs ("clearly", "remarkably", "notably", etc.) that could be assorted with numbers, to reach objectiveness.

We are going to include numbers (i.e. ratio with respect the two densities (pos. and neg.)) in order to be less subjetive.

I. 199: verb missing in sentence.

We are going to rephrase the sentence.

Figures 6 and 7, I. 237: the results that are reported do not say anything about extremes, which are in the tails of the distributions. None of the figures show any change in the tails of distributions. It is already interesting to discuss how the centers of the distributions change.

We agree that the tails do not show significant changes. However, values around +1 (panel b) and -1 (panel d) reveal a shift in the distribution. We discussed these changes as a fingerprint

of wetter or drier conditions. Nevertheless, we agree that the text can be rephrased to emphasize shifts toward more extreme values.

I. 240: why would it be "logical" to extend this study to other ice cores? Would any change (especially for other Greenland ice cores) be expected? If so, this would rather invalidate the whole approach, wouldn't it? As a perspective, what would seem natural (to me), would be to investigate the how natural forcings can affect features of the atmospheric circulation. This issue is barely discussed in the manuscript, while it is a key aspect of paleoclimate studies.

We are going to remove this paragraph. However, Greenland climate variability, recorded by ice core  $\delta^{18}$ O, is influenced by several natural forcings, such as the NAO and AMO. Our point was that ice cores from northern Greenland also reflect atmospheric blocking patterns, while ice cores from other regions of Greenland may capture different local or regional climate signals that are more relevant to those specific locations.